# Nutritional Composition, Phenolic Compounds and Antioxidant Activity of Different Samples of Water Boatmen Eggs (Hemiptera: Corixidae)

**DOI:** 10.3390/foods12010028

**Published:** 2022-12-21

**Authors:** Maria de la Luz Sanchez, Valeria Caltzontzin, Ana A. Feregrino-Pérez

**Affiliations:** Facultad de Ingeniería, Campus Amazcala, Universidad Autónoma de Querétaro, Carretera a Chichimequillas km 1 s/n, El Marques, Queretaro C.P. 76265, Mexico

**Keywords:** aquatic insects, corixids, entomophagy, insect protein, food safety, food information to consumer

## Abstract

The group of aquatic insects collectively called “water boatmen” or “Axayacatl” (Hemiptera: Corixidae) and their eggs, called “Ahuahutle”, have been consumed and cultivated since the pre-Hispanic era in Mexico. Nevertheless, food composition databases contain limited information on the nutritional composition of these eggs. This work evaluates the macronutrients and bioactive compounds of water boatmen eggs obtained from three different locations in Mexico. The primary analyses to be determined for the first time were some bioactive compounds in the eggs, such as phenolic compounds, total flavonoids, condensed tannins content, antioxidant activity (DPPH and ABTS), and, additionally, fatty acids and proximal composition. The results showed that the sample from Hidalgo (AMC) presented the highest number of phenolic compounds (855.12 ± 0.52), followed by ALT (125.52 ± 0.05) and, with the lowest amount, AMT (99.92 ± 0.13), all expressed in an mg GAE/g sample. ALT indicated the highest mol TE/g sample concentration for ABTS (25.34 ± 0.472) and DPPH (39.76 ± 0.054), showing a significant difference in the DPPH method with the AMT samples. The three Corixidae egg samples had between 15 to 18 different fatty acid profiles, and there were statistically significant differences (Student’s *t*-test ≤ 0.05) between the means using MSD. The total fats of the three samples were between 12.5 and 15.5 g/100 g dry basis. In addition, Corixidae eggs are excellent protein sources. Thus, water boatmen’s eggs can be considered to be a food rich in bioactive compounds.

## 1. Introduction

The projected population increase by 2050 of approximately two billion people compared to the current 7.7 billion [1,2] exposes the need to search for alternatives that provide humanity with sustainable protein sources [3]. Edible insects are an alternative source of protein, vitamins, and minerals [4], making them a potential future food [5] and thereby contributing to food security [6]. Although insect consumption (entomophagy) is part of human history [7] and is part of the traditional diet of approximately 2 billion people worldwide [8], their potential as food has not yet been fully exploited, the main reason being a low consumer acceptance [9].

Acceptance of edible insects has focused on studies targeting adults, while acceptance studies targeting the next consumers, such as children, are scarce [10]. The acceptance of edible insects in developed countries is still low, given their preference for animal protein over any other source of origin. In this sense, developing countries, including Mexico, have an inclusive culture, history, and diversity in insect consumption, which is part of the traditional diet in various regions for the general population, including children. Eating insects as a meat substitute provides nutritional and functional advantages with health benefits [11,12]. Although given the diversity of edible insect species, the concentration of nutrients and functional compounds is variable [13,14]. According to Jongema [15], about 2100 insect species are consumed by the populations in developing countries, mainly in the tropics. In these countries, edible insects are collected from nature (trees, plants, agricultural fields, or aquifers), so the diversity, nutritional contribution, and bioactive compounds in terrestrial and aquatic insects are vast.

Mexico has vast biodiversity of insects, with almost 48,000 registered species [16], including aquatic insects. Aquatic insects have been little-exploited as edible insects [17]. These insects spend the egg and larval stages in water, subsequently moving to terrestrial habitats [18,19]. Water boatmen eggs, or Ahuautle (Hemiptera: Corixidae), in pre-Hispanic times were considered a delicacy by the Aztecs; even the Spanish conquerors called them “Mexican caviar” [20]. The eggs are collected from different lakes in the Valley of Mexico, mainly in Lake Texcoco and the waters of Xochimilco; they are a mix of various species of aquatic bugs, such as *Krizousacorixa* spp, *Corisella* spp., *Corixa* spp. (Hemiptera: Corixidae), and *Notonecta* spp. (Hemiptera: Notonectidae). The eggs are 0.5 to 1.00 mm long [21,22]. It is estimated that 3900 millions of waterbug eggs could have been collected in pre-Columbian times in the original area (10,000 ha) of Lake Texcoco; however, the drying of the lake beds and pollution have impacted this practice [23]. Additionally, climate change, greenhouse gases, and acidification of aquifers have generated significant decreases in the quality of the eggs of aquatic bugs (Ahuautle), as well as modifications in the concentration of bioactive compounds. 

Polyphenols have in their chemical structure two phenyl rings and one or more hydroxyl substituents; the structural diversity of flavonoid molecules arises from variations in the hydroxylation pattern and oxidation state of the central pyran ring, resulting in a wide range of bioactive compounds [24]. In recent years, a wide range of spectrophotometric assays have been adopted to measure the antioxidant capacity of foods, the most popular being 2,20-azino-bis-3-ethylbenzthiazoline-6-sulphonic acid (ABTS) and 1,1-diphenyl-2-picrylhydrazyl (DPPH) assay, among others such as oxygen radical absorbance capacity (ORAC) and ferric reducing ability of plasma (FRAP) assay [25]. 

According to Singla et al., polyphenols are natural compounds synthesized exclusively by plants [24] with chemical features related to phenolic substances and eliciting strong antioxidants properties [24]; however, insects have been relatively unexplored as potential sources of natural antioxidants. Many studies suggest that insects might be used as a nutraceutical to alleviate oxidate-induced diseases and as a natural antioxidant additive in the food industry [26], as shown by some of the works that are mentioned below.

Studies such as Yu’s have explored the antioxidant activity in two extracts, ethanolic and water extract, of adult *H. parallela* evaluated with the DPPH method [26]. Another interesting work is that of Chikara et al., which found that the cocoon of *Rondotia menciana* can produce flavonoids that are not present either in the bud or in its food [27]. Other work related to this is that of Ferreres et al., which found that excrements may constitute many bioactive compounds, among them sulfate flavonoids and other flavonoid glycosides that were not detected in the leaves used as food when *Pieris brassicae* were fed [28]. Fu et al. identified over 200 flavonoid metabolites in the larval midgut of *A. pernyi* that come from eight subclasses, namely, flavones (103), flavonols (34), flavonoids (28), flavanones (20), polyphenols (19), isoflavones (9), anthocyanins (9), and proanthocyanidins (4) [29]. Additionally, *Tenebrio molitor* (mealworm) larvae represent one of the most interesting edible insects and could be reared on alternative feeds, such as former foodstuff products. ABTS, DPPH, and FRAP varied significantly as their diet changed. In addition, antioxidant compounds such as δ-tocopherol, γ-tocopherol, and α-tocopherol were identified with significant differences in the different diets [30]. According to the above-mentioned studies, insect metabolites are a source of potentially bioactive compounds, some of which are not found in plants [27,31].

Few nutritional studies have been explored in aquatic insects, as in the Corixidae family, whose knowledge is limited to studies of the proximal composition and fatty acids of both adults and eggs [32,33]. However, polyphenols and antioxidant capacity have yet to be previously explored. In addition, these insects nowadays are not considered for cultivation under controlled conditions and can be an essential food source. Therefore, the nutritional composition of water boatmen’s eggs has not been compared between treatments, as is the case with other insects already being cultivated [30,34,35]. Thus, this work aims to compare the nutritional composition of three samples of Ahuautle (Corixidae eggs) obtained from three locations in Mexico. To our knowledge, this is the first study that provides essential information bases such as the nutritional composition of eggs to promote aquatic insect culturing and consumption through sampling in different locations.

## 2. Materials and Methods

### 2.1. Sample 

The central states in Mexico where Ahuautle consumption has been identified are Michoacan, Hidalgo, and Estado de México [32,33,36,37]. Therefore, different markets were visited in each state to obtain the samples. Likewise, the author interviewed the Ahuautle sellers to obtain information about the origin of the Corixidae eggs; in one case, the sample ALT was collected with the help of a collector in Texcoco lake; this lake has the particularity of being composed of salt water [37]. Table 1 shows a detailed description of the samples obtained in this work.

### 2.2. Proximate Composition

Chemical analyses of three water boatman eggs samples were performed. All values are expressed as the mean of a triplicate determination. The proximate composition was estimated by AOAC (2000) methods: ashes (method no 923.03), proteins (Kjeldahl method, no. 954.01, nitrogen conversion factor: 6.25), and fats (method no 920.39).

Determination of soluble carbohydrates. Total available carbohydrate content of the sample was measured according to the method described by Melo-Ruiz [38], with some modifications based on the Clegg-Anthrone method [39]. The sample (0.1 g) was digested with 5 mL of 2.5 N hydrochloric acid in a water bath at 90 °C for 3 h to hydrolyze disaccharides, trisaccharides, and higher oligomers to their component-reducing sugars. Then, samples were neutralized with calcium carbonate and diluted to 50 mL with distilled water. Following, they were centrifugated, and an aliquot of the supernatant (1 mL) was taken to react with an anthrone (4 mL) in a boiling water bath for 8 min. Anthrone reagent was prepared by dissolving 0.2% (*w*/*v*) anthrone in concentrated sulphuric acid. The absorbance of the reaction mixture was measured at 630 nm against a blank. Dextrose was used to construct a standard curve for quantification, and the results are expressed as mean g/100 g dry basis of sample. All samples were analyzed in triplicate. The caloric value was based on the theoretical energy value of 9 kcal per gram of lipids and 4 kcal per gram of proteins and carbohydrates.

### 2.3. Phenolic Compounds

The extraction was carried out according to the methodology described by Cardador [40]; 200 mg of dry samples were weighed and 10 mL of methanol were added to each sample. They were kept free of light and shaken for 24 h. After this time, they were centrifuged (Thermo Scientific, Waltham, MA, USA) at 5000 rpm for 10 min at 4 °C; the pellet formed at the bottom was eliminated, leaving only the supernatant. Total phenolics were determined using the Folin-Ciocalteu spectrophotometric method [41]. The absorbance was measured at 760 nm, and the results were expressed as the mg equivalent of gallic acid/g of the sample (mg GAE/g sample). Total flavonoids were determined by the Oomah et al. [42] method using rutin as the standard. The absorbance was measured at 404 nm, and the results were expressed as mg equivalent of rutin/g samples (mg RE/g sample). Total condensed tannin samples were evaluated according to the procedure described by Feregrino et al. [43] and modified for use in a microplate using (+) catechin as standard. The absorbance was measured at 492 nm, and the results are expressed as mg of (+) catechin equivalents/g of sample (mg CE/g sample).

### 2.4. Antioxidant Activity (DPPH and ABTS)

The DPPH (2,2-Diphenyl-1-picrylhydrazyl) method was accomplished by Zenil [44]. The analyses were carried out in triplicate. The results were expressed in mg equivalent of trolox/g of sample and absorbance of 520 nm. ABTS (2,2’-azino-bis- (3-ethyl benzothiazolin-6-ammonium sulphonate)) was used to measure antioxidant capacity accomplished by Re [45]. Analyses were measured at 734 nm and in triplicate. The results are in mg equivalent of Trolox/g of a sample.

### 2.5. Content of Total Fats and Fatty Acids by Gas Chromatography–Mass Spectrometry (GC-MS)

GC-MS analysis was carried out as accomplished by Lim [46]. The supernatants of the sample were prepared with the solvent and derivatized. Subsequently, 1 μL of the sample was injected in duplicate in an Agilent gas chromatograph (GC) series 7890A (Wilmington, DE, USA) coupled to a single quadrupole mass spectrometer (MS) detector (Agilent 5975C) equipped with an electron impact (EI) ionization source. The carrier gas (helium) flow rate was maintained at 1 mL/min. The injector temperature was set at 250 °C in splitless mode. An HP-88 capillary column (30 m × 0.25 mm inner diameter × 0.25 μm) was used. The initial oven temperature was 50 °C, which was held for 1 min, and was then raised to 175 °C at 15 °C min^−1^, then raised to 240 °C at 1 °C min^−1^ and held for 5 min. EI energy was set at 70 eV, and the mass range was set at *m*/*z* 50–1100. FAMEs were identified and quantified by comparison with a standard Supelco 37 Component FAME Mix, and data processing was performed using Chem-Station (Agilent Technologies) software version C.01.10.

### 2.6. Statistical Analysis

The data obtained from proximate composition, antioxidant activity, and phenolic compounds content were statistically analyzed. The results of the analyses are reported as the mean with their standard error (SE) using *n* = 3. The results content of total fats and fatty acids were analyzed with the minimum significant difference (MSD) using *n* = 2, and results were reported with mean with their standard deviation (SD). All of them were analyzed with a Student’s *t*-test α ≤ 0.05. Statistical calculation was performed using IBM SPSS Statistics 25.0 (document number 589145; IBM Corporation 2021, Armonk, NY, USA). 

## 3. Results

### 3.1. Proximal Composition

Table 2 compares the composition of macronutrients present in the three Ahuautle samples from different sources. Results show that both ash content (ALT > AMC > AMT) and lipid content (AMT > AMC > ALT) show differences among the three Ahuautle samples, ranging from 5 to 11 and 9 to 14 g/100 g dry basis, respectively, while protein and carbohydrates show differences only in AMC, which had the highest value for protein (74.37 ± 0.186 g/100 g dry basis) and the lowest value for carbohydrates (6.957 ± 0.0964 g/100 g dry basis). The energy contribution (theoretical data) presents a range from 400 to 450 kcal per 100 g dry basis in the samples of Ahuautle, with AMT as the highest (450 kcal/100 g dry basis) and the one with the highest amount of lipids (14.356 ± 0.009 g/100 g dry basis). In comparison, ALT presents a lower energy contribution (400 kcal/100 g dry basis) and a lower amount of lipids (9.195 ± 0.003 g/100 g dry basis).

### 3.2. Phenolic Compounds

Table 3 shows the content of phenolic compounds in a methanolic extract of eggs (Ahuautle). The three Ahuautle samples from different sources contain considerable amounts of total phenols, followed by flavonoids and traces of condensed tannins. The sample from Hidalgo (AMC) presents the highest number of phenolic compounds (855.12 ± 0.52), followed by ALT (125.52 ± 0.05) and, with the lowest amount, AMT (99.92 ± 0.13), all expressed in mg GAE/g sample and indicating a significant difference between the three samples. AMT presents the highest content of flavonoids (2.732 ± 0.1014 mg RE/g sample matter), followed by AMC and ALT (1.698 ± 0.117 and 1.396 ± 0.268 mg RE/g sample matter, respectively), with no significant difference between the AMC and ALT samples. The values of condensed tannins are traces of 0.016 to 0.025 mg EC/g, with AMC presenting a significant difference with the highest value.

### 3.3. Antioxidant Activity

Table 4 shows the antioxidant activity indicated by the three Ahuautle samples analyzed by the ABTS and DPPH methods. The AMC sample shows that the lowest values for both ways (6.34 and 27.44 mol TE/g sample), ABTS and DPPH, respectively, are significantly different from the other two Ahuautle samples analyzed. ALT indicates the highest mol TE/g sample concentration for ABTS (25.34 ± 0.472) and DPPH (39.76 ± 0.054), showing a significant difference in the DPPH method with the AMT samples.

### 3.4. Fatty Acids

Table 5 shows the fatty acid content of three samples of Ahuautle (AMT, AMC, and ALT). The total fats of the three samples are between 12.5 and 15.5 g/100 g dry basis. Ahuautle eggs have between 15 and 18 fatty acids, among them the saturated, monounsaturated, and polyunsaturated types. The most abundant fatty acids in all samples were C16:0 (Palmitic) > C16:1 (Palmitoleic) > C18:1n9t (Elaidic) > C18:2n6t (Linoleic), with significant differences between eggs. In the ALT sample, fatty acids such as C14:1, C15:1, C18:2 (cis), and C20:1 were not detected.

## 4. Discussion

### 4.1. Proximal Composition

Insects are one of the most diverse groups of animals from terrestrial and aquatic ecosystems on the planet. Many insect species are considered edible in their various stages of development (eggs, larvae, pupae, and adults) [47,48]. In addition to having a high nutritional value and according to the World Health Organization (WHO) and Food and Agriculture Organization (FAO), the consumption of insects can be a sustainable strategy in the fight against hunger worldwide [49,50]. The energy contribution of an insect is variable, depending on the species and stage of development. Kinyuru and collaborators [51] estimate a range of 293 to 776 kilocalories (kcal) per 100 g of dry matter. The Ahuautle samples evaluated in this work are within this range (400–450 kcal/100 g dry matter) and present higher values than the 329 kcal/100 g of Ahuautle reported by Rumpold and Schluter [4]. This difference may be attributable to the climatic conditions from which the sample was obtained, storage conditions, and water characteristics. Similarly, the energy intake reported for Ahuautle in this work is higher than other Mexican edible insect species, such as the “thousand-headed snake” (*Latebraria amphipyrioides*) and the grasshopper (*Acrida exaltata*), which provide 349 and 336 kcal per 100 g dry matter, respectively. However, there are also species with higher energy contribution, such as the Mexican moth (*Phassus triangularis*) with 761 kcal per 100 g dry matter. It is essential to highlight that the stage of development influences energy contribution, although most insects are not consumed in their adult stage [4].

The total carbohydrates in the Ahuautle samples evaluated in this work is lower (7 to 7.5 g/100 g dry matter) than the range of 18 to 21 g/100 g dry matter previously reported [4,52]. However, they are not discarded as a source of dietary fiber, and possible nutraceuticals since the carbohydrates in edible insects are mainly provided by the exoskeleton made up of chitin. Chitin is considered the most abundant polysaccharide in nature after cellulose. In insects, it is part of the cuticle and exoskeleton that act as a support element and represents between 11.6 to 137.2 mg per kg of dry matter. This polysaccharide is part of dietary fiber and helps strengthen the human immune system [53].

Moreover, chitosan, a component of active films and packaging with potential applications in food packaging, can be obtained from chitin due to its antimicrobial, antioxidant, and anti-inflammatory properties [54]. The use of insects to obtain chitin has recently increased. A study by Luo [55] observed differences in thermal stability and rheological and morphological characteristics between insect chitosan compared to that obtained from shrimp shells, indicating that these differences contribute to an increase in the application of insect chitosan in sectors such as pharmaceuticals and food. Da Silva [49] suggested that the molts of farm-grown edible insects can be used as a source of chitin and applied in the food industry for their antimicrobial and antifungal properties. 

The amount of protein contained in insects makes them suitable candidates for the development of foods, additives, supplements, and alternative substitutes for protein sources for human and animal consumption, as suggested by the European Commission (EC) and Food and Agriculture Organization (FAO) [8]. The authors of [4,11,52] reported 53 g/100 g dry matter protein for Ahuautle, which is lower than that reported in this work (72 to 74 g/100 g dry matter). In general, insects are mainly composed of protein, representing 38 to 77% of their total mass. As with the rest of the macronutrients in edible insects, the amount of protein varies from species to species, but not in the different stages of their development. On the other hand, the protein quality in edible insects depends on the composition of essential amino acids. In Ahuautle, the presence of Leucine, Phenylalanine, Lysine, Valine, Isoleucine, Threonine, Methionine, and Tryptophan has been reported [52], which are amino acids necessary for the correct development and functionality of the body and, therefore, consuming Ahuautle complies with the amino acid content of the diet recommended by the WHO. It is considered to be a possible solution to the nutritional deficiencies that affect developing countries [56].

Lipids are the second principal component in insects; additionally, in the larval and egg stage, the concentration of lipids is higher. The concentration of lipids in the samples of Ahuautle analyzed in this work is shown in a range of 9 to 14 g/100 g dry matter. Melo [52] report a lipid concentration of 4.33 g/100 g dry basis, differences that may be attributable to various factors such as place of origin, measurement methods, and storage, as well as extrinsic factors such as feeding and rearing conditions [4,57]. Based on the literature, the lipids present in edible insects are constituted in more significant proportion by unsaturated fatty acids than saturated fatty acids, a ratio that varies depending on the species and diet of the insect [58].

### 4.2. Phenolic Compounds

The composition of insects does not only comprise lipids, proteins, carbohydrates, and minerals. Insects provide other organic compounds such as vitamins, nucleic acids, and phenols [59]. Phenolic compounds are phytochemical compounds whose origin is generally vegetable. They possess diverse biological activities such as antioxidant, anticancer, antimicrobial, hypotensive, and antihyperglycemic, to mention a few. Some of the above biological activities have been reported in edible insects and their by-products; however, there are few reports on the presence of phenolic compounds in edible insects.

Chantawannakul [60] reported the presence of vitamins, terpenoids, polyphenols, sulfur compounds, and glycosides in edible insects. All of them have biological properties that contribute to the health of humans and animals that can consume them. Therefore, edible insects can be considered nutraceutical foods. DeFelice [61] can define nutraceutical as follows: “Food containing natural compounds or substances that provide nutritional and functional benefits to the organism”. The presence of polyphenols in insects has not been clearly documented. There are reports of insect-derived products containing polyphenols, such as honey or propolis [62,63].

Baigts and collaborators [64] reported the presence of chlorogenic acid, epicatechin, coumaric acid, caffeic acid, gallic acid, kaempferol, and epicatechin gallate in *Ascra cordifera*, *Brachygastra mellifica*, and *Hermetia illucens*. They were identified by liquid chromatography–electrospray ionization–tandem mass spectrometry (LC-ESI-MS/MS). In our study, we indicated the presence of total phenols in a range of 100 to 850 mg GAE/g of samples. Therefore, this is the first report on the presence of phenolic compounds in Ahuautle samples. As Baigts and collaborators [64] reported for other edible insect species, the Crixidae egg samples analyzed in this work also indicate the presence of flavonoids as mg catechin equivalents. Catechin is one of the most abundant types of flavonoids present in vegetables and has also been reported in chocolate, red wines, and various types of tea, among other foods. Flavonoids have considerable biological activities, such as antioxidant, antimicrobial, anti-inflammatory, and anticancer activities, among others [60]. Insects cannot synthesize flavonoids but can accumulate them by their diet type [65]. Flavonoids provide insects with pigmentation [66], chemical defense [67], and visual communication [68], and play a role in mating and repelling predators [69]. On the other hand, the report of the presence of condensed tannins in this work is the first report that we are aware of both for the samples of Corixidae eggs analyzed and for other species of edible insects.

### 4.3. Antioxidant Activity

Edible insects have been consumed not only for culinary and cultural reasons; many edible insects are consumed for their biological properties, including antioxidant activity. Antioxidant activity is the ability of compounds in food to neutralize and/or eliminate reactive oxygen species (ROS) through various reaction mechanisms [70]. A research study [71] compiled several data where the antioxidant capacity of edible insects is evaluated in both in vivo and in vitro models. The most used in vitro methods are FRAP, DPPH, and ABTS+, with *Tenebrio molitor*, *Acheta domesticus*, *Gryllodes sigillatus*, *Bombyx mori*, *Hermitia illucens*, and *Lethocerus indicus* as the most studied edible insects in relation to their antioxidant capacity. It should be noted that the present work reports the first known record of antioxidant activity in Ahuautle through the antiradical mechanism by the DPPH and ABTS+ methods. The analyzed Ahuautle samples from three different points of origin indicate a superior DPPH radical scavenging power (27.44 to 39.76 mol TE/g sample) compared to the ABTS+ radical scavenging capacity (6.34 to 25.34 mol TE/g sample), showing statistical differences between the samples evaluated in both methods.

Antioxidant activity is generally related to the presence of polyphenolic compounds, which are also significantly present in the Ahuautle samples analyzed. Zhang and collaborators [72] indicated a positive correlation between the content of phenolic compounds and antioxidant capacity in an insect-based tea used in China as a traditional medicine for the last 100 years. Unfortunately, studies on edible insects and polyphenols are scarce. The antioxidant activity reported in various insects has been related to their high protein content, i.e., the presence of biopeptides. Biopeptides are small protein molecules composed of less than 20 amino acid residues and weighing less than 6 kDa. Biopeptides have various biological properties, including antioxidants [73]. Another study carried out by Vercruysse et al. [74] documented that biopeptides in the cotton leafworm (*Spodoptera littoralis*) present high ferric ion-reducing antioxidant power (FRAP) in addition to presenting DPPH radical scavenging capacity. Di Mattia et al. [75] reported that grasshoppers, silkworms, and crickets present four times more antioxidant power than orange juice. Mudd et al. [76] indicated that edible crickets (*Gryllodes sigillatus*) contain peptides with antioxidant properties and can be used as a functional food to combat stress.

### 4.4. Fatty Acids

Fatty acid composition in three Ahuautle samples was analyzed in this work to evaluate the potential of these eggs as a source of bioactive lipids. A high content of saturated and monounsaturated fatty acids is present in the samples; however, among the polyunsaturated fatty acids, the levels of C20:4, C20:5, and C22:6 (DHA) acids stand out, all of which have been proven to have a positive effect on human health. The monounsaturated fatty acids of the aquatic insect eggs (Hemíptera: *Corixidae* and *Notonectidae* spp.) evaluated in this work are mostly the same as the terrestrial edible insects previously reported [77]. However, the difference is that the eggs have Trans- C18:1, and the terrestrial insects instead have Cis-C18:1.

According to Twining [78], aquatic insects are rich in omega-3 fatty acids, and the results of this work show significant differences in these fatty acids among the samples. Results suggest that the diet of aquatic insects affects the content of fatty acids. Other work that supports this affirmation is Dennis’s work [79], which concludes that adding a source of n-3 fatty acids to insect diets can improve the nutritional quality of three insects (*Hermetia illucens*, *Acheta domesticus*, and *Alphitobius diaperinus*).

The results obtained in this work support the statement of Hilaire [80], which states that the fatty acid composition of insects can vary greatly, even within the same species, depending on their feeding. These substrates in aquatic life contain larvae and algae with traces of polyunsaturated fatty acids [80]. The presence of polyunsaturated fatty acids and other fatty acids in aquatic insects is associated with diet and enzymatic activity [81]. Aquatic insects have enzymes such as D5 desaturase and D6 desaturase that can synthesize polyunsaturated fatty acids [82,83].

Oleic acid is dominant in terrestrial insects. However, the same is not valid for aquatic insects. In our study, the AMC sample was the only one that presented 0.10 ± 0.01 of cis-oleic acid, while the AMT and ALT samples were not detected. Therefore, we can affirm that the diet of aquatic Hemiptera (Corixids) directly affects the fatty acid composition of their eggs (Ahuautle). However, we still know little about the absorption of lipids by insect oocytes [84], and studies still need to analyze the diet of the corixids that lay eggs (Ahuautle). Hence, with other insects, this assertion has been observed [85,86].

There are two primary sources of trans fats: natural and industrial. For the first, animals and their products, such as cows, sheep, goats, milk, and butter, have natural trans fats; the second source of trans fatty acids is the industrial process of hydrogenation of vegetable oils [87]. This last source of industrial trans fatty acids is directly related to obesity, insulin resistance, cardiovascular disease, cancer, and inflammation [88,89]. However, Pipoyan and collaborators reviewed damages caused by trans fatty acids in natural and industrial products. They concluded that industrial fatty acids are more harmful than natural ones produced by ruminants [87]. Considering that food with natural trans fatty acids cannot be removed, Table 5 shows the most abundant fatty acids in three samples of Ahuautle, reported as g/100 g dry basis. The range values are 3.3–4.3 to C16:0 (Palmitic) > 2.5–2.6 to C16:1 (Palmitoleic) > 1.3–2.0 to C18:1n9t (Elaidic) > 0.9–1.6 to C18:2n6t (Linolelaidic), with significant differences between eggs. However, the average content in this work of trans fatty acids (Palmitoleic and Elaidic) is around 1.4 g/100 g dry basis, which is already within the 14.4 g/100 g dry basis of total fats. Therefore, the quantity of trans fatty acids is minimal and should not be of concern to the consumer due to all that Corixidae eggs’ nutritional value provides. The World Health Organization (WHO) and the Pan American Health Organization (PAHO) recommend the elimination of industrially produced trans fatty acids to prevent noncommunicable diseases, such as coronary heart disease [90]. Likewise, The World Health Organization recommended in 2003 that all trans fats be limited to less than 1% of overall energy intake. The EFSA’s opinion in 2004 suggested that foods in the EU intended for consumers must contain less than 2 g of industrial trans fatty acids per 100 g of fat [91].

Aquatic insects have an average protein content of 59.55%, which is higher than that of conventional animal meats [78,92]. Generally, the protein content in insects can vary according to the growth stage of the insect; for example, the content in silkworm pupae is 15.8%, while in beef meat it is 21.35%, chicken meat, 21.30%, and pork meat, 19.40% [93]. Considering that the recommended daily allowance for adults is 0.66 g/kg/d, a minimal quantity of insect protein covers the requirements. Furthermore, investigations identified that proteins from aquatic insects not only contain 45.93–62.01% essential amino acids, but also have a good balance of different kinds of amino acids [94]. Meanwhile, the ratio of essential amino acids in aquatic insect proteins is close to protein requirements for humans, indicating a high nutritional value of aquatic insects [95]. In contrast with the role of plant feeders that most terrestrial edible insect species play, most aquatic edible insects are carnivorous animals. Besides the differences in physiology and metabolism, there are differences in fat, fatty acid, amino acid, and mineral element contents between terrestrial and aquatic insects [96]. Compared with terrestrial insects, there is very little information available on the nutritional value of aquatic insects. However, it is know that insects are high in mineral content (e.g., iron and zinc), B vitamins, and essential amino acids [17]. 

Given that over 2000 insect species are eaten around the world, edible aquatic insects account for about 15% of the total number. The species belong to eight orders. Among them, Coleoptera, Odonata, and Hemiptera contribute over 3/4 of the number of species, and they are all predatory. Since the quality of the water in which insects develop affects their composition, it is not recommended to consume large quantities of wild insects, as they are usually found in places with heavy metals, pesticide residues, and uric acid, which does not ensure their safety. For the large consumption of aquatic insects, it is better to standardize their rearing before they can be safely eaten [97,98].

To our knowledge, this is the first study demonstrating how the wild diet of aquatic insects may modify the nutritional composition of their eggs; in addition, it provides essential information, for the first time, about their nutritional composition, phenolic compounds, and antioxidant activity. This study aims to promote the consumption of aquatic insects. Accepting the water boatmen’s eggs in international gastronomy is good, mainly because of their flavor, which is related to dried shrimp or caviar used in international gourmet dishes [99]. Besides the fact that the rearing of aquatic insects has not been explored, it would be convenient to start their exploration to sustain the eggs of the corixids. In general, future emphasis needs to be put on designing rearing protocols so that the bulk production success achieved with crickets, mealworms, and flies can be replicated for aquatic insects [17].

## 5. Conclusions

Aquatic insects are an important source of protein that can be a sustainable alternative to combat world hunger. The feed, origin, and stage of development influence the composition and nutraceutical contribution of aquatic insects. Ahuautle is a food that provides macrocompounds that contribute to nutrition but also contains bioactive compounds such as phenolic compounds that provide antioxidant activity and, thus, health benefits. The compounds in Corixidae eggs can be an ecological alternative to modulate oxidative stress and a source of molecules with other biological activities that contribute to the prevention of chronic diseases. However, more studies are still needed to confirm the efficacy of these edible insects in this area. The fatty acid composition of insects can vary greatly, even in their eggs, depending on their feeding substrates. These substrates in aquatic life could contain feeding with polyunsaturated fatty acids. The amount of protein in Corixidae eggs in this work (72 to 74 g/100 g dry matter) is a possible solution to nutritional deficiencies. About 90% of the fatty acids contained in Ahuautle are beneficial to health. On the other hand, phenolic compounds stand out in this study, followed by antioxidant activity (ABTS and DPPH). Thus, Corixidae eggs can be considered to be a food rich in bioactive compounds.

## Figures and Tables

**Table 1 foods-12-00028-t001:** Locations of samples of Corixidae eggs.

Sample Name	Sample Description
AMC *	The sample AMC was obtained commercially in a market of the state of Hidalgo, Mexico. According to the seller, the collector obtained the eggs in a lake in Hidalgo.
AMT *	The sample AMT was obtained commercially in a market of the state of Michoacan, Mexico. According to the seller, the collector obtained the eggs in a lake in Michoacan.
ALT	The sample was obtained from the Texcoco lake in Estado de Mexico. The ALT sample is particular because the water of Texcoco Lake is salty. An aquatic insect collector gathered the eggs in the coordinates 19°32′12.4′′ N 99°00′9.9′′ W.

* The collectors from the states of Michoacan and Hidalgo did not provide the sampling coordinates of the samples. However, they confirmed that they belong to a lake in the above locations.

**Table 2 foods-12-00028-t002:** Macronutrients in three different samples of corixidae eggs (g/100 g dry basis).

Ahuautle Sample	Ashes	Fat	Proteins	Carbohydrates	Caloric Value (kcal/100 g)
AMT	5.485 ± 0.226 ^C^	14.356 ± 0.009 ^A^	72.641 ± 0.128 ^B^	7.516 ± 0.060 ^A^	450
AMC	7.798 ± 0.643 ^B^	11.523 ± 0.223 ^B^	74.37 ± 0.186 ^A^	6.957 ± 0.0964 ^B^	430
ALT	10.97 ± 0.017 ^A^	9.195 ± 0.003 ^C^	72.47 ± 0.126 ^B^	7.364 ± 0.082 ^A^	400

Analyses were carried out in triplicate, and the table shows the mean with SD. Means with different letters in the same column are statistically different (Student’s *t*-test α ≤ 0.05) Protein by Kjeldahl NX6.25. Kcal = theoretical data. AMC: sample of Corixidae eggs from Hidalgo, Mexico; AMT: sample of Corixidae eggs from Michoacan, Mexico; ALT: sample of Corixidae eggs from Estado de Mexico (Texcoco lake).

**Table 3 foods-12-00028-t003:** Phenolic compounds content in Corixidae eggs using methanol extracts.

Ahuautle Sample	Phenolic mg GAE/g Sample	Flavonoids mg RE/g Sample	Condensed Tannins mg CE/g Sample
AMT	99.92 ± 0.13 ^C^	2.732 ± 0.1014 ^A^	0.01614 ± 0.003 ^B^
AMC	855.12 ± 0.52 ^A^	1.698 ± 0.117 ^B^	0.0251 ± 0.0034 ^A^
ALT	125.52 ± 0.05 ^B^	1.396 ± 0.268 ^B^	0.01594 ± 0.0038 ^B^

mg GAE/g sample (mg Gallic acid equivalents/g sample); mg RE/g sample (mg equivalent of rutin/g sample); mg CE/g sample (mg of (+) catechin equivalents/g of sample). The average represents the value of 3 repetitions. Means with different letters in the same column are statistically different (Student’s *t*-test α ≤ 0.05). AMC: sample of Corixidae eggs from Hidalgo, Mexico; AMT: sample of Corixidae eggs from Michoacan, Mexico; ALT: sample of Corixidae eggs from Estado de Mexico (Texcoco lake).

**Table 4 foods-12-00028-t004:** Antioxidant activity of Corixidae eggs determined by ABTS and DPPH measured as μmol TE (Trolox Equivalent/g sample).

Ahuautle Sample	ABTS μmol TE/g Sample	DPPH μmol TE/g Sample
AMT	25.28 ± 0.429 ^A^	35.22 ± 0.305 ^B^
AMC	6.34 ± 0.257 ^B^	27.44 ± 0.178 ^C^
ALT	25.34 ± 0.472 ^A^	39.76 ± 0.054 ^A^

Results are means of triplicate analysis ± standard error of the mean (SEM). Samples that do not share the same letter are significantly different (Student’s *t*-test α ≤ 0.05). AMC: sample of Corixidae eggs from Hidalgo, Mexico; AMT: sample of Corixidae eggs from Michoacan, Mexico; ALT: sample of Corixidae eggs from Estado de Mexico (Texcoco lake).

**Table 5 foods-12-00028-t005:** Content of total fats and fatty acids in different Corixidae egg samples (g/100 g dry basis).

	AMT	AMC	ALT
C14:0 (Myristic)	0.33 ± 0.02 ^A^	0.25 ± 0.001 ^B^	0.14 ± 0.0003 ^C^
C14:1 (Myristoleic)	Nd	0.38 ± 0.06 ^A^	Nd
C15:0 (Pentadecanoic)	0.14 ± 0.01 ^A^	0.12 ± 0.003 ^B^	0.09 ± 0.002 ^C^
C15:1 (*cis*-10-Pentadecanoic)	0.10 ± 0.01 ^A^	0.09 ± 0.02 ^A^	Nd
C16:0 (Palmitic)	3.44 ± 0.16 ^C^	4.33 ± 0.004 ^A^	4.07 ± 0.01 ^B^
C16:1 (Palmitoleic)	2.67 ± 0.14 ^A^	2.61 ± 0.003 ^A^	2.17 ± 0.03 ^B^
C17:0 (Heptadecanoic)	0.34 ± 0.003 ^C^	0.51 ± 0.008 ^A^	0.47 ± 0.006 ^B^
C17:1 (*cis*-10-Heptadecenoic)	0.26 ± 0.005 ^B^	0.37 ± 0.04 ^A^	0.27 ± 0.005 ^B^
C18:0 (Stearic)	1.09 ± 0.03 ^B^	1.38 ± 0.004 ^A^	0.50 ± 0.03 ^C^
C18:1n9t (Elaidic)	1.49 ± 0.06 ^B^	1.35 ± 0.04 ^C^	2.01 ± 0.01 ^A^
C18:2n6c (Linoleic)	Nd	0.10 ± 0.01 ^A^	Nd
C18:2n6t (Linolelaidic)	0.99 ± 0.03 ^C^	1.62 ± 0.02 ^A^	1.31 ± 0.01 ^B^
C18:3n6 (γ-Linolenic)	0.07 ± 0.01 ^C^	1.23 ± 0.04 ^B^	1.78 ± 0.03 ^A^
C18:3n3 (α-Linolenic)	1.06 ± 0.05 ^A^	0.03 ± 0.01 ^C^	0.33 ± 0.004 ^B^
C20:1n9 (*cis*-11-Eicosenoic)	0.13 ± 0.01 ^A^	Nd	Nd
C20:3n3 (*cis*-11, 14, 17-Eicosatrienoic)	Nd	0.07 ± 0.0001 ^B^	0.08 ± 0.001 ^A^
C20:4n6 (Arachidonic)	0.17 ± 0.01 ^C^	0.43 ± 0.008 ^B^	0.68 ± 0.02 ^A^
C20:5n3 (*cis*-5, 8, 11,14, 17-Eicosapentaenoic)	0.76 ± 0.04 ^B^	0.35 ± 0.12 ^C^	0.86 ± 0.02 ^A^
C22:6n3 (*cis*-4, 7, 10, 13, 16, 19-Docosahexaenoic)	0.11 ± 0.004 ^A^	0.03 ± 0.004 ^B^	0.14 ± 0.06 ^A^
Total fats	13.15 ± 0.59 ^B^	15.18 ± 0.34 ^A^	14.90 ± 0.06 ^A^

Nd = no detected. Results are means of duplicate analysis ± standard deviation (SD). The minimum significant difference (MSD) was analyzed, and different letters in the same row correspond to statistically significant differences (Student’s *t*-test α ≤ 0.05) among the different egg samples. AMC: sample of Corixidae eggs from Hidalgo, Mexico; AMT: sample of Corixidae eggs from Michoacan, Mexico; ALT: sample of Corixidae eggs from Estado de Mexico (Texcoco lake).

## Data Availability

The data supporting this research are in the computer database of the laboratory of metabolites and nanocomposites at the Universidad Autónoma de Querétaro, campus Aeropuerto.

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
