# Peer review of "Nutritional Composition, Phenolic Compounds and Antioxidant Activity of Different Samples of Water Boatmen Eggs (Hemiptera: Corixidae)"

_foods, 2022, doi:10.3390/foods12010028_

Round 1
Reviewer 1 Report
The article entitled “A comparative study on the antioxidant activity and fatty acids in “Ahuautle“” by Luz Sanchez M et al., studying the proximate composition, antioxidant activity and the fatty acid profile of three Ahuautle sample. The article is interesting, well worked and contains complex information regarding the intended purpose. However, there some changes required, as reported below:
- a graphical abstract would be very helpful for better understanding of the article. Please insert one.
-The introduction part can be improved: please write some general information about polyphenols (https://doi.org/10.3390/molecules27041345). Write a detailed description of phenol, flavonoid compounds what can be found in the material analyzed.
-The part of material and methods: proximate composition: please write in more detail the determination of carbohydrates and energy value consists of
- please change in table 1: instead of kilocalories, write energy value
-Result and discussion: it is well written and a comparison was made between the results obtained by others
Author Response
Dear reviewer
The authors are grateful and appreciate all comments from the reviewers, which undoubtedly helped to improve the manuscript. In the text, the authors highlight in yellow the reply concerning each suggestion made by the reviewer.
A document is attached detailing point-by-point the observations and modifications made to the manuscript.

Reviewer 2 Report
attached

Author Response
Dear Reviewer
The authors are grateful and appreciate all comments from the reviewers, which undoubtedly helped to improve the manuscript. In the text, the authors highlight in yellow the reply concerning each suggestion made by the reviewer.
A document is attached where each of the observations is broken down point by point.

Reviewer 3 Report
The study on the edible value of aquatic insects provides an important scientific basis for the development of edible insect species and the alternative protein sources for human in the future. Therefore, the author's research work is very significant. I have some questions about the manuscript for the author.
1. The title of the manuscript is not quite appropriate. The authors did not conduct a comparative study of nutrient composition between different species or with known edible insects. At the same time, this paper covers a wide range of nutrients, not limited to antioxidant activities and fatty acids. Please do not use the common name of local insect in the title instead of the scientific name.
2. The three acronyms (AMC, AMT, ALT) that appear throughout the article do not specify their meaning. Do these three treatments refer to different sampling sites, or different species? If the three treatments refer to different sampling sites, the authors need to identify the specific species of aquatic insect eggs in the three treatments and describe them in table.
3. The background and methods in the abstract are too redundant and could be simplified.
4. In Introduction, the author never stated what the purpose of the comparative study was.
5. If the author intends to promote the edible value of aquatic insects, then the nutritional value should be analyzed according to the specific species of aquatic insects collected, and compared with known edible insects and common meat protein source animals.
6. Line 117, The author does not elaborate the specific statistical analysis method clearly. Mean analysis is a big category of statistical analysis, including many statistical analysis methods, such as variance analysis, T test and so on.
7. In Discussion, the author needs to add more content about how eating aquatic insects is beneficial to human beings and how to exploit and utilize them.
Author Response

(The authors gave the same response as above.)

Round 2
Reviewer 1 Report
Thank you for the answers. The mentioned modifications was done. I propose that the article be published!
Author Response
Thank you very much for your observations, we appreciate the feedback and enrichment of our manuscript.
Reviewer 3 Report
The author has supplemented and modified the content of the manuscript, so the quality of the article has been greatly improved.
In the article, it would be nice if the authors could provide specific species of the water boatmen eggs from different sampling sites. However, this does not affect the quality of the article. This study provides theoretical guidance for the development of edible functions of aquatic insects.
Author Response
We really appreciate your comment. Unfortunately, the term Ahuautle is used to describe a mixture of eggs from different aquatic insects, highlighting Corixidae and Natonectidae, both of the genus Hemiptera, which makes it difficult to focus on a single type of insect. However, we do not rule out the possibility of carrying out further research in order to obtain the characteristics and nutritional contribution of the egg of a single species.